:ᐧᔤ: PLOS | ONE

# Examining influential factors for acknowledgements classification using supervised learning

**Min Song** [1]*, **Keun Young Kang**[1], **Tatsawan Timakum**[1,2], **Xinyuan Zhang**[3]

**1** Department of Library and Information Science, Yonsei University, Seoul, Korea, **2** Department of Information Sciences, Chiang Mai Rajabhat University, Chiang Mai, Thailand, **3** School of Information Management, Wuhan University, Hubei, China

* min.song@yonsei.ac.kr

## Abstract

Acknowledgements have been examined as important elements in measuring the contributions to and intellectual debts of a scientific publication. Unlike previous studies that were limited in the scope of analysis and manual examination. The present study aimed to conduct the automatic classification of acknowledgements on a large scale of data. To this end, we first created a training dataset for acknowledgements classification by sampling the acknowledgements sections from the entire PubMed Central database. Second, we adopted various supervised learning algorithms to examine which algorithm performed best in what condition. In addition, we observed the factors affecting classification performance. We investigated the effects of the following three main aspects: classification algorithms, categories, and text representations. The CNN+Doc2Vec algorithm achieved the highest performance of 93.58% accuracy in the original dataset and 87.93% in the converted dataset. The experimental results indicated that the characteristics of categories and sentence patterns influenced the performance of classification. Most of the classifiers performed better on the categories of financial, peer interactive communication, and technical support compared to other classes.

## Introduction

Acknowledgements have increased their presence to become a routine part of scientific publications, and they have played their own role in the scholarly communication process since the 1960s [1]. Generally, acknowledgements are used to convey appreciation to diverse scientific entities, such as funding bodies, referees, and research participants [2]. The content of acknowledgements sections vary across papers, since no uniform writing protocol like those for citation and authorship exists. In addition, the unstructured text as well as limited data availability both make it difficult to analyze the acknowledgements section in an automatic way. Multiple schemes have been proposed to classify this different acknowledgements content, ranging from just looking at one type to examining more classes relying on a manual supply of acknowledgements data at the early stage. In 2008, the Web of Science (WoS) database began to include funding fields, which opened up new possibilities for data mining and for

**Data Availability Statement:** The datasets underlying this study are available publicly via two DOIs 1) Acknowledgement training dataset https://doi.org/10.6084/m9.figshare.11321852 and 2) Acknowledgement classification model (Doc2Vec

Classifier Model) https://doi.org/10.6084/m9.figshare.11330564. Also, all files are available from http://informatics.yonsei.ac.kr/acknowledgement_classification/

**Funding:** This study was supported by the National Research Foundation of Korea (NRF-2019R1A2C2002577). The funders had no role in study design, data collection and analysis, decision to publish, or preparation of the manuscript.

**Competing interests:** The authors have declared that no competing interests exist.

analyzing the information contained in the acknowledgement section of papers in the same way as citation and authorship. Paul-Hus et al. [3] provide a detailed description of WoS funding information. The Scopus database began to provide funding metadata but no access to the full text in 2013 [4]. Recently, some databases, publishers, and open access repositories have included the full text of the acknowledgement section for scientific papers.

The recent increased availability of full text makes it possible to analyze the characteristics of acknowledgements in a profound manner, and the sheer volume of acknowledgements calls for a kind of automatic classification. The early scholars of acknowledgements classification applied handcrafted statistical approaches to classify them, and they presented evidence of the intellectual influences and social interactions that lie behind the authorship of research. Examples include analyzing funding [3, 5] and finding the network and co-authorship network [6–7]. Others include examining the categories and patterns of acknowledgements in each discipline [8–9] and investigating the acknowledgement behavior [10]. However, the previous methods are time-consuming and are not applicable to the analysis of large amounts of data. In addition, if we can detect the hidden knowledge in the contents of intellectual indebtedness, we better understand why and how authors appreciate their collaborators. In the field of content analysis, most researchers have focused on examining abstract as well as full-text, which are structured content, by applying several methods such as co-authorship and citation analysis for bibliometrics related research [11–14]. In contrast, the text of acknowledgement remains unexplored. Thus, the present study can give an insight into the unusual scientific resources, such as syntax services, experiment supports, or other research contributors mentioned in acknowledgements section. This is likely to further present research opportunities for future studies.

Recently, machine learning as well as deep learning techniques have been widely used for bibliometric research as an effective tool to rank the scientific research [15], predict further citation [16–17], classify citation motivation [18], and author disambiguation [19]. However, these machine learning and deep learning techniques have not been applied to automatic acknowledgements classification at PubMed scale.

The main goal of the present study was to develop the automatic classification method for the acknowledgements section. In particular, we aimed at two objectives. Firstly, we aimed to create a large-scale training dataset for the automatic classification of acknowledgements. Secondly, we examined which factors affected classification performance. We investigated the effects of the following three aspects: classification algorithms, types of category, and text representations. For classification algorithms, we adopted several supervised learning algorithms, and we compared the performance in terms of accuracy, precision, recall, and F1-score for each classifier. For the effects of categories, we classified our acknowledgements into six categories and examined the characteristics of each category that may affect the performance. For the effects of text representations, we converted a sentence into a simplified format by utilizing several features and applied the Named Entity Recognition (NER) model with Stanford CoreNLP's seven classes for feature generation to examine the impact of text representations on the performance of classifiers. The present study serves as a core base for analyzing the entire collection of acknowledgements sections from full-text repositories like PMC and understanding the characteristics and patterns of acknowledgements on a large scale. To facilitate research of acknowledgements section, we have made both the training dataset and acknowledgements classification model publicly available at http://informatics.yonsei.ac.kr/acknowledgement_classification/acknowledgement_classification.html.

The main contribution of the present study is to explore the most effective and automatic classification of acknowledgements by selecting salient features for classification and the state-of-the-art algorithms.

The rest of the paper is organized as follows. Section 2 provides a summary of the related work on acknowledgements classification and automatic citation classification. The details of methods are presented in Section 3 including building training data process which explains how we created the categories of acknowledgements and classified them manually. Section 4 is the experiment results which presents the classification algorithms we used, effects of classification algorithms, effects of text representation, and effects of categories. We conclude our study in Section 5.

## Related work

### Acknowledgements classification

Acknowledgements have become important paratext of research as a source of authorship and contributorship information. This research trend has drawn increasing attention from scholars, who have studied and pointed out the different acknowledgements patterns. For instance, McCain [9] studied bibliometric data and classified the types of acknowledgements research-related information. In the same year, Cronin [8] explored the typologies of acknowledgements in *JASIST*. Tiew and Sen [20] classified the acknowledged individuals in the *Journal of Natural Rubber Research*. Rattan [21] categorized the acknowledgements in the *Annals of Library and Information Studies* and classified the acknowledgements appearing in the DESI-DOC Journal of Library & Information Technology by applying eight-layer topologies [22]. The categories of the different scholars are presented in Table 1 (A1). However, the most common types of acknowledgements in scientific literature are summarized from previous studies in Table 1 (A2), and these include Access, Peer Interactive Communication, Moral, Technical, Clerical, Financial, Manuscript and Editorial, and Unclassifiable Support.

Prior works on acknowledgements analysis, the scholars applied approximately statistic approaches to examine the acknowledgements, which presents the evidence of intellectual influences and social interactions that lie behind the authorship of research. However, with its growing massive number of scientific literatures as well as the change of knowledge domains, the role of associations have been described increasingly in the acknowledgements [27–28]. This change challenges us to explore the hidden information from the enormous of acknowledgements by developing appropriate data mining approaches to classify the content of them.

### Automatic citation classification

Citation classification is the one approach of content-based citation analysis that aims to understand the influence of scientific literature on authors, such as the author's purpose regarding citing a paper or the function of a citation. However, identifying the key concepts in citation context automatically is complex and they are difficult to classify, because the differences among categories are too subtle and many categories make it hard to apply automatic methods, especially using linguistic markers [29–30]. Citation classification with three classes regularly refers to citation polarity [31]. It differs from citation function classification, which uses different schemes and categories for specific knowledge fields that annotate a corpus from the literature to train a model for automatic classification due to specific academic expressions in different fields linked to technical terms [32].

Consequently, to define categories of citation function, many methods require human classification and annotation to distinguish several types of criteria as well as the integration of additional information (e.g., citation location and frequency) with machine learning based classifiers [33], such as technical words and sentiment words of previous knowledge [32], to overcome the abovementioned problem. For example, Small [34] applied supervised machine learning and corpus linguistics to distinguish citation contexts of biomedical papers classified

**Table 1. Acknowledgements categories of previous studies.**

| (A1) Acknowledgements categories of different scholars | | (A2) Acknowledgements categories of previous studies in conclusion |
|---|---|---|
| Mackintosh [23] | 1. Facilities<br>2. Access to data<br>3. Help of individuals | *1. Access support* relates to access to research information and infrastructure facilities (*Access to data*, *Access to research-related information*, *Access to unpublished data*, and *Access*) |
| Patel [24] | 1. Technical assistance<br>2. Theoretical support | *2. Peer interactive communication (PIC) support* relates to people who provided related research advice, valuable suggestions, intellectual guidance, critical reviews, comments, discussions, and assessments (*Conceptual*, *Theoretical support*, and *Trusted assessor*) |
| McCain [9] | 1. Access to research-related information<br>2. Access to unpublished data<br>2. Peer interactive communication<br>4. Technical assistance<br>5. Manuscript preparation | *3. Moral support* relates to those who encouraged and supported access to facilities, those who gave permissions, and dedication from family or friends (*Help of individuals and Prime mover*)<br>*4. Technical support* relates to help in technical expertise, technology, laboratory equipment and sample preparation, programming and experimental techniques, and study and analysis design (*Technical*, *Technical expertise*, *Instrumental/technical*, and *Technical assistance*) |
| Cronin [8] | 1. Paymaster<br>2. Moral support<br>3. Dogsbody<br>4. Technical<br>5. Prime mover<br>6. Trusted assessor | *5. Clerical support* relates to secretarial services from colleagues such as routine data acquisition and management (*Dogsbody*)<br>*6. Financial support* relates to grants/scholarships from external or internal funding sources (*Paymaster and Financial*)<br>*7. Manuscript and editorial support* relates to help preparing a manuscript, editing, proofreading, language translating, critical reviews, and comments (*Manuscript preparation and Editorial/linguistic support*) |
| Tiew and Sen [20] | 1. Moral support<br>2. Financial support<br>3. Access<br>4. Clerical support<br>5. Technical support<br>6. Peer interactive communication<br>7. Unclassifiable | *8. Unclassifiable* relates to those types that cannot be classified according to the categories above. |
| Cronin, Shaw, And LaBarre [25–26] | 1. Conceptual<br>2. Editorial<br>3. Financial<br>4. Instrumental/technical<br>5. Technical expertise<br>6. Moral<br>7. Unknown | |
| Rattan [21] | 1. Access support<br>2. Moral support<br>3. Financial support<br>4. Technical support<br>5. Peer interactive communication support<br>6. Clerical support<br>7. Editorial/linguistic support<br>8. Unclassifiable | |

by type of method and non-methods to discover linguistic and other cue words that indicated how the papers were perceived in the citing passages.

Ding et al. [29] explained that there are two main methods to organize categories of citation text: using a rule based on cue words or phrases set in a decision tree classification to classify extracted citations and applying machine learning techniques to create different classifiers. Some existing studies of automatic citation classification are presented below.

Teufel [35] employed rhetorical multi-classification by labeling sentences based on their functional roles into seven categories. Later, Teufel et al. [36–37] utilized supervised machine learning (kNN) to classify citation function automatically and used linguistic features such as cue phrases, verb and voice, location, and human annotation. Similar to the multi-classification task, Angrosh et al. [38] studied the problem of an imbalance of multi-classification, which makes annotation difficult. They limited labeled data by annotating every sentence in

the related work section to tackle an annotation scheme problem. They used generalization terms (lexicon), ensuring the previous sentence had citations as a feature, and applied a Conditional Random Field (CRF) as a classifier.

Moreover, Hirohata et al. [39] used a supervised machine learning approach to classify the sentences in abstracts into four classes, including objectives, methods, results, and conclusions, by employing CRFs, n-gram and sentence location as a feature. The experimental results showed that CRFs shaped the rhetorical structure of abstracts better than n-gram. Dong and Schäfer [40] used linguistic feature sets to define a citation classification schema with four classes—background, fundamental idea, technical basis, and comparison—by applying supervised learning classifiers (Sequential minimal optimization (SMO), BayesNet, naïve Bayes, J48, and Random Forest). They named each sentence to the corresponding section category. The results proved that the feature set with the part-of-speech (POS) tags and added syntactic patterns was most effective. Hernández-Alvarez et al. [30] proposed that citations should be categorized into four dimensions—function, polarity, aspects, and influence—by using the influence and impact of the words interchangeably. They developed a method for a classification scheme and manually annotated a corpus that was traced with meaningful keywords and labels to help identify the specific functions. They applied SMO to train a support vector machine (SVM) and used naïve Bayes and a simple bag-of-words baseline classification. The result confirmed that using four classes and adding keywords and labeling specific functions minimized the error of analysis.

To identify the important features for rhetorical classification, Widyantoro and Amin [13] explored an effective method for citation sentence extraction and classified the rhetorical categories of citation sentences. They performed the analysis by applying feature vectors of citation sentence containing term frequency, sentence length, thematic word, and cue phrase feature groups and employed an additional extraction rule. They classified citation sentences as belonging to the problem, method, and conclusion rhetorical categories using naïve Bayes, complement naïve Bayes, and decision tree. The experimental results showed that the thematic word feature outperformed the other feature groups and that the decision tree and naïve Bayes classifiers performed comparably well.

From the previous studies, we have considered the various factors for building the training dataset and generated features for classification algorithms that may affect automatic classification.

## Methods

It is pivotal to make a training dataset that properly represents acknowledgements and to develop a classification algorithm best suited to acknowledgements. To this end, we reviewed the previous studies of citation classification and applied their strategies to create the training dataset. In addition, we explored several different sentence representations to see which one worked best for acknowledgements classification. The overall procedure of the proposed approach is presented in Fig 1.

The methods applied to the experiments consisted of three parts:1) data collection and pre-processing, 2) dataset preparation and transformation, and 3) algorithm implementation and analysis. For data collection and pre-processing, we used the PubMed Central dataset to extract the acknowledgements section. After collecting and pre-processing, we built the training set for classification in two ways. The first training set is comprised of original sentences along with the classification label manually assigned by the experts. The second set consists of the converted sentences by a regular expression and Named Entity Recognition (NER) techniques. The reason for providing two different sets of training data is to examine which

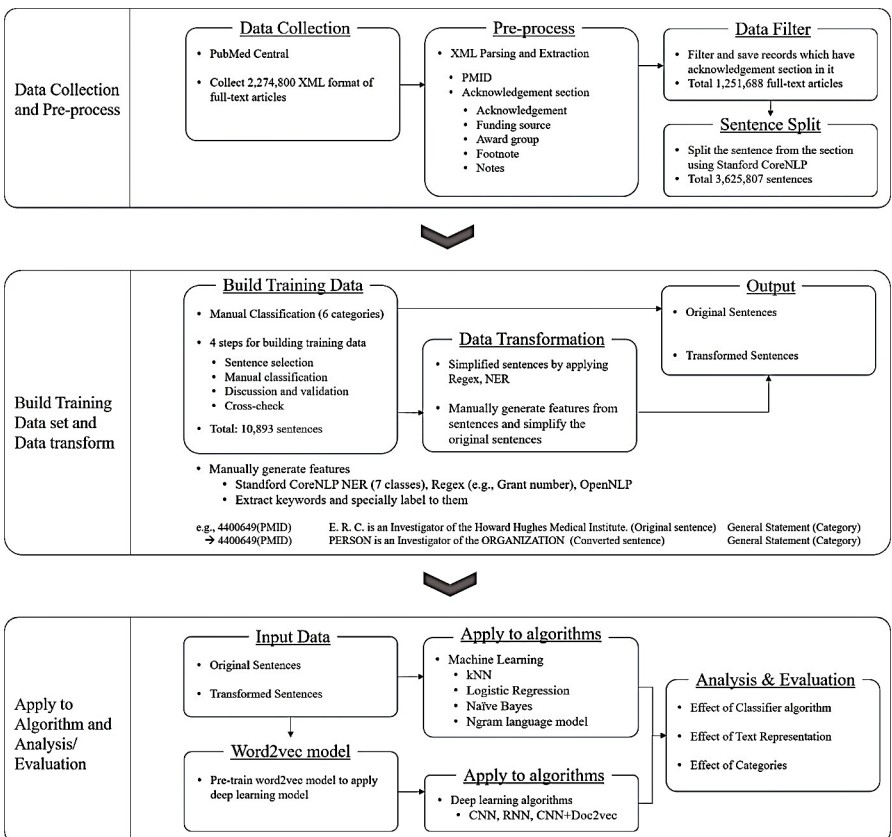

**Fig 1. Overall procedure of proposed approach.**

training dataset works better as features for automatic classification. With these two different training sets, we applied various deep learning and machine learning algorithms to measure the performance of the classification of acknowledgement sentences. In the analysis phase, we examine the impact of classification algorithms as well as the types of text representation on the performance of classification.

## Data collection

We downloaded the entire full-text collection from PubMed Central database (PMC) ftp://ftp.ncbi.nlm.nih.gov/pub/pmc. We collected approximately 2,274,800 full-text articles in XML format, parsed the XML format of the articles to text format, and extracted the PMID and acknowledgements section. When parsing the XML data, we extracted several sections related with acknowledgements: 1) acknowledgements, 2) funding source, 3) award group, 4) footnotes, and 5) notes. In total, 1,251,699 articles had acknowledgement-related sections.

## Preprocessing

We preprocessed the articles into sentences. The sections consisted of sentences with different intentions. Therefore, when we classified the intention of or reason for the acknowledgement, we had to split the section paragraphs into sentences. Otherwise, the acknowledgements paragraph or section itself could not be classified accurately. Then, we separated 1,251,699 articles

into 3,625,807 sentences to classify the sentences into several intentions of acknowledgement. This means that each acknowledgements section contained about 2.89 sentences on average.

The reason for separating an acknowledgements section into sentences is because there could be multiple reasons for acknowledging someone or some entities. Thus, to analyze types of acknowledgements at a fine-grained level, we decided to split the acknowledgements section into sentences.

## Building training data

The three researchers who are Ph.D. candidates majoring in the library and information science participated in making the training set. To make the training set useful and reliable, we selected 10,893 sentences randomly from the whole dataset. The training set was built using the following four steps: (1) Sentence selection, (2) Manual classification, (3) Discussion and validation, and (4) Crosschecking with each other.

**Step 1: Sentence selection.** As preparation for building the training dataset, we randomly selected 10,893 sentences from 3,625,807 sentences manually. These sentences were used as the base of the pre-trained model in the deep learning algorithm.

**Step 2: Manual classification.** To classify the sentences of acknowledgement, we considered the content, language patterns, and schemes based on previous studies and the characteristics of our raw data of acknowledgements. In this study, we divided the classes of acknowledgements into six categories, which are shown in Table 2.

We devised new class labels that were not introduced in the prior studies to cover the content of the data that we classified, such as a category of "Presentation" and "Peer interactive communication and technical support". For some categories, we still borrowed the class labels from the previous studies such as "Financial" [20, 22, 25–26].

**Table 2. Categories of acknowledgements in this study.**

| Categories | Description |
|---|---|
| **1. Peer interactive communication and technical support** | The acknowledgement statement is about a person or organization for advising and supporting during the research process and the reporting of the study as a manuscript. This category represents specific suggestions and technical and analysis support in research, such as study design, providing help in the use of laboratory tools and space, and sample preparation. Moreover, it includes discussions, comments, and assessments on the study, report, and manuscript, including reviewing, editing, proofreading, and linguistic support. |
| **2. Declaration** | The declaration statement relates to conflicts of interest, access to unpublished data, and copyright as well as the authority of a person or organization. It also relates to ethics approval and consent with permission to publish the data and manuscript. This category involves moral support and privacy concerns as well as declarations of lack of funding. |
| **3. Presentation** | The statement of presentation states whether a portion of the work, including an abstract, poster, or oral presentation, was presented at a conference, proceeding, or seminar. |
| **4. Financial** | This acknowledges any grants or scholarships received by the researcher from external or internal funding. |
| **5. General acknowledgement** | In this statement, the authors express gratitude to persons or organizations that encouraged them during the study. This acknowledgement is not of specific assistance. |
| **6. General statement** | The general statement presents information not related to a study directly (e.g., information on a person or institute). This is an example of the *unclassifiable* category. |

To determine the categories of acknowledgements, we used the keywords and their usage patterns in the acknowledgements section. A similar approach was adopted in the study of Paul-Hua et al. [41] that we used as a guideline to categorize the acknowledgements sentences. They extracted acknowledgements indexed in the WoS, SCI-E, and SSCI to identify the types of acknowledgements. They applied linguistic pre-processing tools, such as a tokenizer and POS tagger, and modified the grammatical rule set for noun phrase chunking to obtain a result. We took this result to consider our classification as well. For example, to consider our class of *Peer interactive communication and technical support*, related terms such as *discussion*, *advice*, *guidance*, *image analysis*, *and laboratory tools* were observed for classification. The summary of noun phrases is shown in Table 3.

Furthermore, we also considered other features such as technical terms, thematic words, numerical information, and context in each sentence. Moreover, we focused on the grammar structure such as verb tense and voice as well as the pattern of the sentences. For example, "This work was funded by two NIH MIDAS grants (U01GM110712 and 5U54GM111274) and a WHO grant 353558 TSA 2014/485861-0." was classified into the financial category by considering grammar (*was funded by* followed by *organization name*), financial terms (*grant*), and the number of grants.

Another example is "Bi was responsible for the conception, design, data acquisition, analysis, drafting, and revising of the article and the final approval of all versions of the article." We categorized this sentence into the peer interactive communication and technical support category because "responsible for" was followed by analysis and technical support terms. We also classified this sentence that contained "responsible for" into the declaration category.

To classify acknowledgements sentences manually, we provided more details on our rules in a guideline (S1 File) located at https://doi.org/10.6084/m9.figshare.11302280. The guideline (nine pages, six categories) describes the categories with examples and provides decision aids in semantically ambiguous cases. After developing our acknowledgements categories, three researchers manually classified the 10,893 sentences to create the training dataset.

**Table 3. Examples of noun phrase patterns in each type of acknowledgements category.**

| Acknowledgements categories with noun phrase patterns of previous study Paul-Hua et al. [41] | Acknowledgements categories with noun phrase patterns of this study (Six categories) |
|---|---|
| **Technical support** <br> *image, equipment, computational resource, measurement, code, calculation* <br> **Peer discussion/communication and intellectual debt (reviewers, editors, and colleagues)** <br> *fruitful discussion, helpful discussion, valuable discussion, guidance, feedback, valuable suggestion, useful comment, helpful comment, valuable comment, insightful comment, editor, reviewer, anonymous reviewer, anonymous referee* <br> **Funding** <br> *financial assistance, financial support, partial financial support, funder, grant sponsor, fund, fellowship, studentship, recipient analysis, data collection* <br> **Funding and disclosure of conflict of interest** <br> *fee, honorarium, conflict of interest, consultant, employee, financial conflict, financial interest, financial involvement, sponsor, other relevant affiliation* <br> **Participation of human subject** <br> *family, patient, participant, participation, trial* <br> ***Performing of research*** <br> *preparation, technical assistance, excellent technical assistance* | **1. Peer interactive communication and technical support** <br> *discussion, comment, suggestion, advice, guidance, editing, editor, reviewer, meta-analysis, image analysis, microarray analysis, laboratory tools, study design* <br> **2. Declaration** <br> *ethics approval, consent, human subject, approval, declare, disclosure, compliance, copyright, authorization, dedicate* <br> **3. Presentation** <br> *presentation, present, poster, oral, conference, proceeding, seminar, annual, symposium, annual meeting* <br> **4. Financial** <br> *funded by, supported by, financial support, personal fee, invest in, supported in part by grant, funds, scholarships, agency, fellowship, salaries, personal fees* <br> **5. General acknowledgement** <br> *research center, university, laboratory, lab, participants, volunteers, patients* <br> **6. General statement** <br> verb to be + *nouns* <br> *professor, researcher, doctoral fellow* |

**Step 3: Discussion and validation.** After manual classification, the three researchers discussed the categories and sentences that were hard to classify (e.g., multiple categories in one sentence). In the discussion session, they shared, added, and edited descriptions/guidelines about each category. In addition, we extracted and shared some keywords in the sentences that represented the categories. Sentences can be represented with keywords, so we fed the transformed sentences into deep learning algorithms to see if they performed better than the sentences without transformation. In the validation process, each student checked the training set again based on the discussed guidelines.

**Step 4: Crosschecking with each other.** Before making the final version of the training set, we performed a secondary step for checking the data. Three researchers who majored in the library and information science crosschecked all of the data with the guidelines. For this step, we made agreements and calculated the agreement rate as follows:

1. If all three researchers classified the sentence into the same category, the agreement rate was 100%.

2. If two researchers classified it into the same category and one was different, the rate was ~ 66.67%.

3. If all researchers classified it differently but discussed and decided to focus on one category, the rate was 0%.

According to these rules, we made the final version of the training dataset with an average agreement rate of 97.27%, and the final number of sentences for each category, shown in Table 4, was 1,815 on average and 10,893 sentences in a total of the training dataset.

In the evaluation of any machine learning or deep learning algorithms, the training dataset is split into two sets at 80:20 rate for the training and the validation set, respectively.

## Results

### Classification algorithms

For classification algorithms, we employed state-of-the-art classification algorithms, including statistical representation-based document classification algorithms, such as k-nearest neighbors (kNN), logistic regression, naïve Bayes, and the n-gram language model, and neural representation-based classification algorithms, such as convolutional neural network (CNN), recurrent neural network (RNN), and CNN combined with Doc2Vec.

Before applying deep learning algorithms to acknowledgements content, we built an acknowledgements-specific word2vec model. When we built the word representation model as a pre-trained model for training contextual factors in a sentence and detecting contextual similarity between words, we applied pre-trained word vectors in the CNN and RNN

**Table 4. Number of sentences for each category.**

| Category | Number of Sentences |
|---|---|
| Declaration | 1,871 |
| Financial | 2,450 |
| Peer Interactive Communication and Technical Support | 2,532 |
| Presentation | 1,209 |
| General Acknowledgement | 1,448 |
| General Statement | 1,383 |
| **Total** | **10,893** |

processes. For this, we used 1,000 acknowledgements sentences in the training process with a layer size of 100 and a window size of 20 to improve contextual understanding. This word2vec model, which was fitted for the acknowledgements dataset, helped reflect the sequence of words (contextual information) in the deep learning process. We applied classification algorithms as follows.

**CNN combined with Doc2vec.** In this model, we applied the Doc2vec model [42–43] as an input feature for CNN to overcome the inefficient zero-padding problem when using CNN with raw text. We transformed raw text into sentences through embedding by Doc2Vec to represent sentences with linguistic as well as semantic properties and rules. The assumption of this algorithm is that Doc2Vec makes similar document vector representations within the same category. When training with this model, we firstly made the input for the Doc2vec model as a vector format based on the original form of sentences with dimensionality of 500, a learning rate of 0.025, a batch size of 1000, and 20 epochs. The input was then applied to the CNN combined Doc2vec model. In the model building process, we set the layer size to 500 with a fully connected multi-dense layer and one softmax output layer.

**RNN.** The RNN is an extension model of the convolutional feedforward neural network. It learns sequential data within the sophisticated recurrent hidden layer, which can be suited to natural language text [44–45]. When applying the RNN, we split the data into training and test datasets in advance and then used the training dataset for our RNN model. In the setting, we had one LSTM layer with a softmax activation function and one softmax output layer with a MCXENT loss function. For hyper-parameters, we used a batch size of 100 and 1000 epochs.

**CNN.** The CNN consists of a multiple convolutional layer, a pooling layer, and a fully connected layer. It is best suited to train two-dimensional data, and it is usually applied for image recognition [46]. Here, we applied Kim's architecture [47]: three convolutional layers with multiple filter widths and 100 feature maps, one max pooling layer, and a fully connected layer with a softmax output layer. For parameters, we used a batch size of 200, a vector size of 100 (specific to the acknowledgements word2vec model), and 3,000 epochs.

**kNN.** The kNN is the simplest non-parametric machine learning algorithm that can classify unclassified test data to some label of the nearest sample based on the training set [48–49]. For the kNN, we applied LingPipe API, setting $k$ (the number of categories) to 6, and evaluated the accuracy (i.e., how well the data was classified).

**Logistic regression.** The logistic regression model is commonly used in statistics, and it is a form of binomial regression [50]. Recently, it has usually been applied as a multinomial logistic regression form to solve multi-class classification problems [51–52]. We also applied LingPipe API for the logistic regression, and for the setting, we set the learning rate to 0.002, base of the exponent to 0.9975, minimum epochs to 100, and maximum epochs to 1,000.

**Naïve Bayes.** Naïve Bayes, known as a simple Bayesian algorithm, is a machine learning algorithm that is commonly and effectively used for text categorization [53–54]. The algorithm is sensitive to feature selection, and the feature value is independent of the features among a given class of variables [48, 55]. The naïve Bayes model applied in this paper involves the bounded character language model. We used the default setting in LingPipe API; for instance, we trained the boundary character n-gram model, and it processed the data in 6-character sequences as the default n-gram size of LingPipe API.

**N-gram language model.** At first, it was used for language modeling in speech recognition and machine translation, but it has also been applied to text classifiers [56–57]. In the case of the n-gram language model, we used character n-gram models in LingPipe API. We also used the default setting with the 6-character sequence length in the model.

We compared the performance of these seven algorithms on two datasets of acknowledgements sentences, original raw text data and converted data, each of which was labeled with six

categories (declaration, financial, presentation, peer interactive communication and technical support, general acknowledgement, general statement). The labeled data with 10,893 sentences were processed for training with various algorithms. We conducted an evaluation of acknowledgements classification by examining the effects of three important aspects of factors affecting classification performance: classification algorithms, categories, and text representations.

## Effects of classification algorithms

With regard to the performance difference among the seven classifiers, we summarized the overall performance of the classifiers by accuracy, precision, recall, and F-measure, and the results are reported in Fig 2.

Overall, the assessment results indicate that to classify the acknowledgements sentences, CNN+Doc2Vec achieves the highest performance on all measures for the original dataset: accuracy (93.58%), precision (93.68%), recall (93.82%), and F-measure (93.74%). For the converted dataset, CNN+Doc2Vec also achieves the best performance on all measures but only slightly better than logistic regression, naïve Bayes, and the n-gram language model for accuracy (87.93%), precision (88.17%), recall (88.57%), and F-measure (88.36%). We provide a detailed analysis of the performance difference in classifiers on original raw text representations vs. on transformed representations in the section below.

In contrast, the RNN scores the lowest in both datasets for accuracy (55.80%), precision (61.91%), recall (54.66%), and F-measure (55.91%). With the transformed dataset, the performance of the RNN for accuracy, precision, recall, and F-measure is 69.03%, 71.64%, 70.07%, and 70.54%, respectively.

However, the effectiveness of the CNN+Doc2Vec and RNN algorithms of the different datasets shows that the weighted average scores of CNN+Doc2Vec performance decreases slightly when applied to the converted dataset, which, opposite with RNN, could improve the performance compared to the original dataset. However, the performance of the other algorithms is slightly different between the two datasets.

## Effects of text representation

**Text representation.** We examined whether different text representations influence the performance of classifiers. If so, we further examined how different the performance was and what factors affected the performance difference. For the converted text representation, the overall procedure is illustrated in Fig 3.

As shown in Fig 3, we converted a sentence into a simplified format with several features. For feature generation, we applied the NER model with Stanford CoreNLP's seven classes, including person, organization, location, date, money, percent, and time [58]. We also applied pattern matching for grant number and keyword matching. In addition, we applied OpenNLP's chunker and ReVERB [59] to extract subject-predicate-object tuples. Table 5 presents examples of original and converted sentences for each category.

The original dataset has an average of 20.7 tokens per sentence. The maximum number of tokens is 91, and the minimum number is 2. Overall, 93.9% of sentences include fewer than 40 tokens (Fig 4).

However, the converted dataset has an average of 10.6 tokens per sentence. The maximum number of tokens is 55 and the minimum is 1. In the converted dataset, approximately 97.3% of sentences contain fewer than 20 tokens (Fig 5).

**Performance comparison between datasets.** Throughout the series of experiments, the original dataset and the converted dataset show a clear difference among algorithms (Table 6).

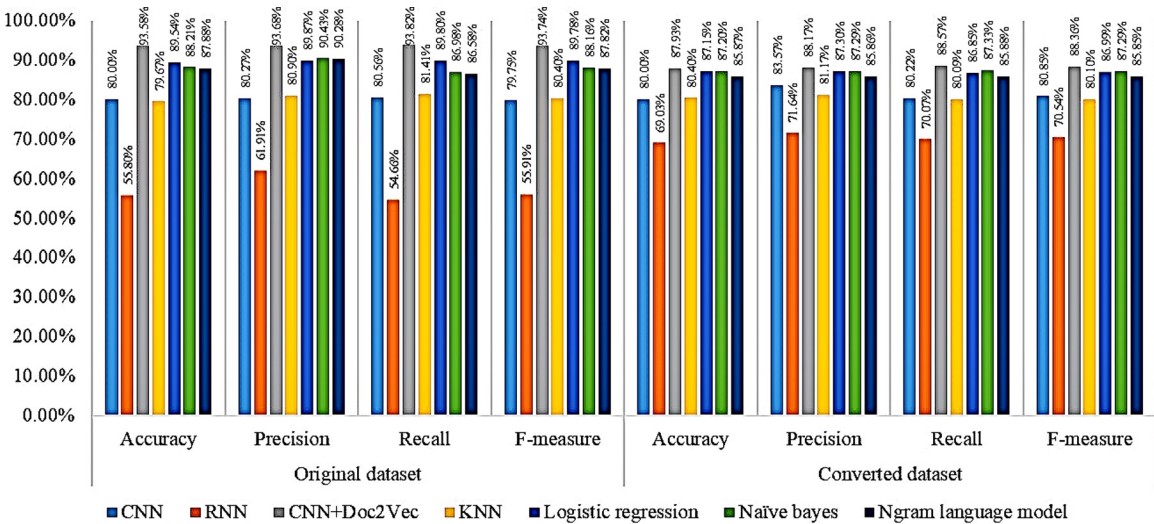

**Fig 2. Performance results of seven classifiers.**

### Input File & Shuffle

- PMID / Sentence / Label

- 5958345 Zoher Gueroui and Christophe Sandt are gratefully acknowledged for access to the fluorescence and FTIR microscopes respectively. Peer interactive communication and technical support

### Pattern and Keyword Match

- Pattern match
  - Grant Number

- Extracted keyword match
  - Peer interactive communication and Technical support
  - Finance
  - Declaration
  - Acknowledge
  - Presentation

### S/P/O Extraction

- Extract Subject-Predicate-Object from replaced sentence.

- OpenNLP: Chunking → ReVERB
- ReVERB: Subject - relation - Object

- Replaced sentence
PERSON and PERSON are gratefully acknowledged for access

### NER

- Stanford CoreNLP's 7 classes

- {*Zoher Gueroui*=PERSON, *FTIR*=ORGANIZATION, *Christophe Sandt*=PERSON}

- Replaced sentence
PERSON and PERSON are gratefully acknowledged for access to the fluorescence and ORGANIZATION microscopes respectively .

### Output

- Result

5958345 *PERSON and PERSON are gratefully acknowledged for access PERSON PERSON acknowledgment acknowledgment* Peer interactive communication and technical support

**Fig 3. Overall procedure of text conversion.**

**Table 5. Examples of text representations.**

| Category | Original sentence | | Transformed sentence |
|---|---|---|---|
| Declaration | All participants gave written informed consent before inclusion after adequate explanation of the study protocol. | | All participants gave written informed consent before inclusion declaration |
| Financial | ChemMatCARS Sector 15 is supported by the National Science Foundation under grant number NSF/CHE-1346572. | | This project has been supported by the ORGANIZATION GRANT_NO finance |
| Peer Interactive Communication and Technical Support | Zoher Guerouai and Christophe Sandt are gratefully acknowledged for access to the fluorescence and FTIR microscopes, respectively. | >> | PERSON and PERSON are gratefully acknowledged for access |
| Presentation | This work was presented in part at the 104th Annual Meeting of the American Society for Microbiology, New Orleans, LA, 23–27 May 2004. | | This work was presented in part SET LOCATION ORGANIZATION GRANT_NO presentation |
| General Acknowledgement | We appreciate the participation of the patients and their families. | | We appreciate the participation of the patients |
| General Statement | R. N. is an Investigator of the Howard Hughes Medical Institute. | | PERSON is an Investigator of the ORGANIZATION |

The performance is different by F1-measure, ranging from 0.2% to 15% in all algorithms with which we experimented.

In the CNN and RNN results, the converted dataset performs better than the original dataset. In the RNN, the gap between the two different datasets is almost 14.6% in F1-measure. The performance result of the RNN with the original dataset is comparatively low (55.9%). It may be attributed to the characteristics of acknowledgements datasets and the RNN's long-term dependency problem. The dataset itself consists of a large volume of vocabulary, and its sentence lengths vary. On the other hand, the converted dataset is simplified through a dimension reduction process, and the word vector variation is reduced. Due to these reasons, the original dataset is not appropriate for the RNN.

While the performance gap of the RNN is large between different text representations, the CNN has a 1.1% difference in F1-measure between the two datasets. The CNN sets the sentence length as equal, and this can cause the zero-padding problem. This is the point that leads to the difference from the CNN combined with Doc2Vec model. Zero padding is ineffective,

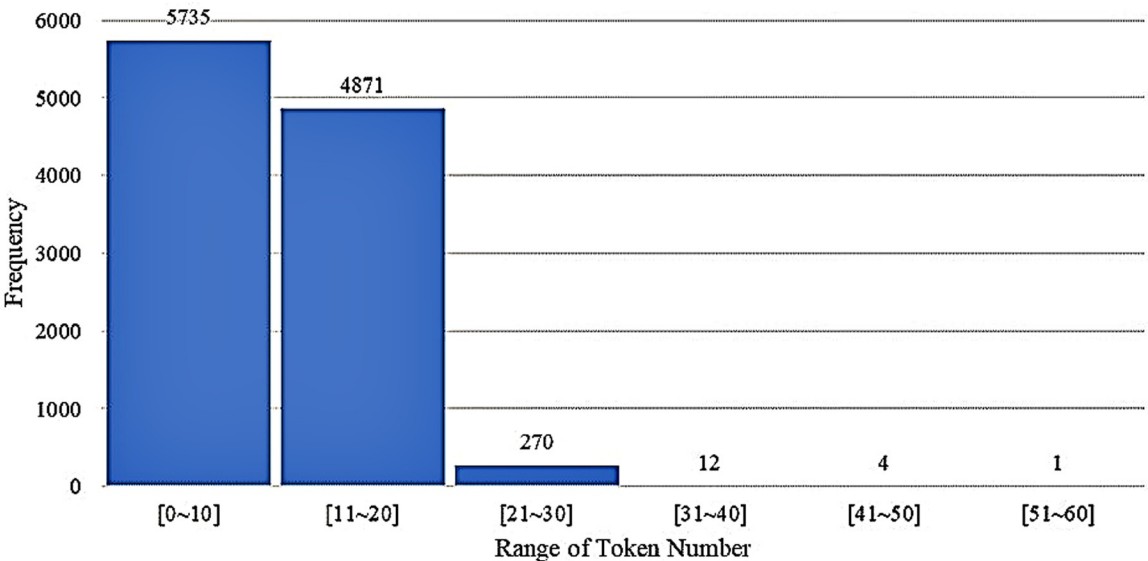

**Fig 4. Frequency distribution for original dataset.**

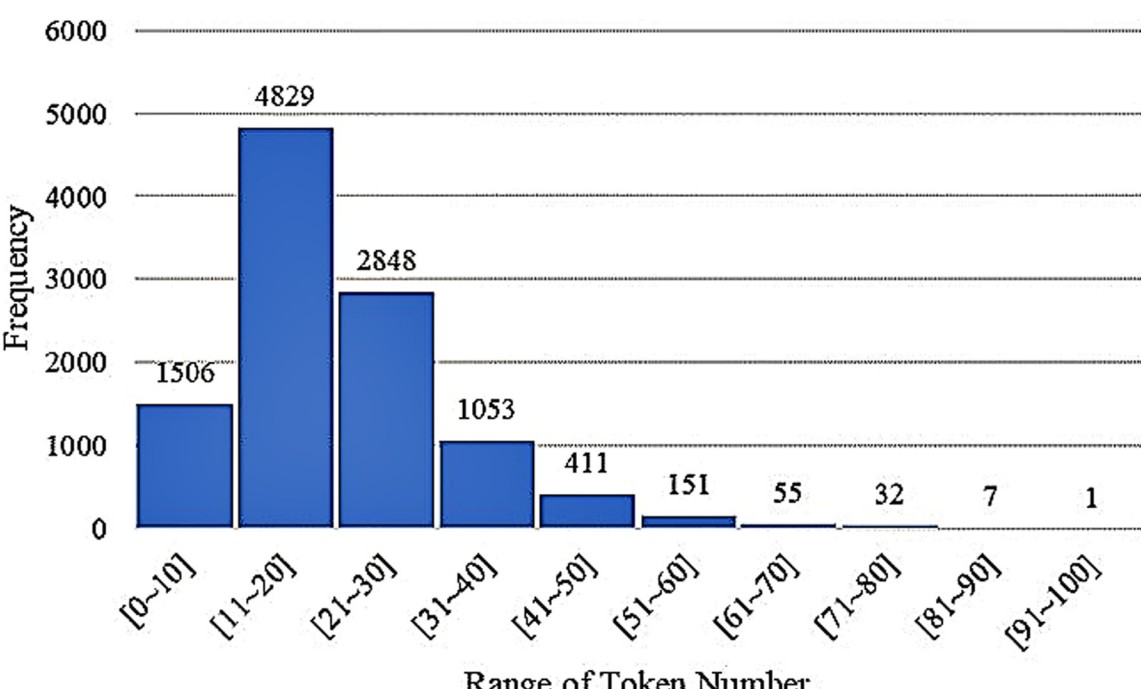

**Fig 5. Frequency distribution for converted dataset.**

but it can help a model to learn without sentence length variation. In addition, NER-processed sentences have simplified patterns. Thus, this is why the CNN performs better with the converted dataset than with the original dataset (e.g., CNN: 1.1%, RNN: 14.6%). However, the CNN combined with Doc2Vec algorithm performs 5.4% better with the original dataset. Without considering the number of tokens in the document, Doc2Vec represents a document itself as a vector and transforms text as well as category label simultaneously. Therefore, it can learn contextual information in a document. The original dataset conveys rich contextual terms, and this is why the CNN combined with Doc2Vec can learn better using this dataset.

Overall, machine learning algorithms perform on average 0.97% better with the original dataset in F1-measure (Table 6). This means that the data conversion process does not positively affect the performance. The performance of these algorithms is better than that of deep learning-based ones, except CNN combined with Doc2Vec.

**Table 6. Performance result of classification algorithms (F1-measure).**

| Model | Original dataset (%) | Converted dataset (%) |
|---|---|---|
| CNN | 79.7 | 80.8 |
| RNN | 55.9 | 70.5 |
| CNN + Doc2vec | **93.7** | **88.3** |
| kNN | 80.3 | 80.1 |
| Logistic regression | 89.7 | 86.9 |
| Naïve Bayes | 88.1 | 87.2 |
| N-gram language model | 87.8 | 85.8 |

## Effects of categories

Here, we report the results of the characteristics of each category that may affect the performance of algorithms. To label the training dataset for classification, we referred to the guidelines as mentioned in the previous section (S1 File). The language patterns and lengths of sentences of each category differed. We calculated the lengths of sentences in our dataset, and the results showed that of all 10,893 sentences, the average sentence length was 20.7 tokens. Moreover, some categories could be distinguished by specific key terms, noun phrases, or sentence patterns. The characteristics and the examples of sentences in each category are shown in detail in S2 File located at https://doi.org/10.6084/m9.figshare.11302289. The following summarizes the characteristics of each category.

1. *Peer interactive communication and technical support*. The average sentence length is 22.3 tokens. The names of persons and organizations, technical terms, and verbs describing support or assistance are used often. Compound sentences are usually found in this category, which increases the sentence length.

2. *Declaration*. The average length is 18.7 tokens. Most of the sentences are short and simple with explicit terms and verbs that state the role in research.

3. *Presentation*. The average length is 25.8 tokens. This class does not contain expressions of appreciation. Most of the sentences are long but not compound sentences; the length comes from conference names, locations, and dates. In addition, it consists of presentation terms that are easy to distinguish, such as poster and oral presentation.

4. *Financial*. The average length is 22.0 tokens. The statements are mostly simple sentences. The sentences are both short and long, but the long sentences contain explicit financial terms and verbs associated with grant support. Therefore, the acknowledgements content of this category is easy to distinguish.

5. *General acknowledgement*. The average length is 19.2 tokens. These are moral support statements. Consequently, there are more informal terms in the sentences than technical terms. The names of persons and organizations are mentioned; however, pronouns and nouns such as patients, staff, and participants are often used instead of personal names.

6. *General statement*. The average length is 15.9 tokens. The sentence pattern is simple, the length is short, and mostly verbs are used. The names of persons and organizations are often found in this class.

The effects of using different classifiers on each category are reported by accuracy measures with the original dataset in Table 7.

Table 7 shows that the financial category is the best suited for most of the classifiers except for the CNN and RNN. The experimental results also indicate that the financial category achieves the highest accuracy at 17.57% on average of all seven algorithms. The second best category is peer interactive communication and technical support, which works well with the five classifiers except for the CNN and RNN. The category of general statement showed the lowest performance from many classifiers at 10.61%, which is slightly different from the general acknowledgements category, which showed 11.11% accuracy on average of the seven algorithms.

These results seem to indicate that the classifiers can distinguish the type of class easily from specific financial noun phrases and grant number information, which occurred frequently in acknowledgements statements. However, the general acknowledgement and general

**Table 7. Average of proportional performance results on categories with original dataset.**

| Algorithms (Accuracy) | Categories | | | | | |
|---|---|---|---|---|---|---|
| | Peer interactive communication and technical support | Declaration | Presentation | Financial | General acknowledgement | General statement |
| CNN (80.00%) | 8.57 | 17.14 | 11.43 | 7.14 | **21.43** | 14.29 |
| RNN (55.80%) | 5.48 | 8.27 | **19.50** | 12.57 | 6.07 | 3.92 |
| CNN+Doc2vec (93.58%) | 21.66 | 17.21 | 9.73 | **21.98** | 11.38 | 11.61 |
| kNN (79.67%) | 14.04 | 13.77 | 11.01 | **18.31** | 10.65 | 11.89 |
| Logistic regression (89.54%) | 18.72 | 15.37 | 11.84 | **20.88** | 10.60 | 12.12 |
| Naïve Bayes (88.21%) | 20.28 | 15.97 | 11.75 | **21.02** | 8.86 | 10.33 |
| N-gram language model (87.88%) | 20.24 | 15.92 | 11.75 | **21.06** | 8.81 | 10.10 |
| *Average of performance per category* | *15.57* | *14.81* | *12.43* | *17.57* | *11.11* | *10.61* |

statement categories contain various general terms that are difficult for algorithms to predict randomly.

General acknowledgement and general statement show the highest performance for the CNN at 21.43% and 14.29%, respectively. These results confirm that the CNN performs well with categories that are less complex and contain short text, since these two categories contain simple sentences and few technical terms (e.g., "AB is a KUL postdoctoral fellow."). However, CNN+Doc2Vec outperforms the other classifiers in the long and compound sentences as well as the sentences with technical terms or verbs, such as the declaration category, which used specific verbs to state the role in a study (e.g., to declare, to be responsible for, and to not be involved). We believe that this performance comes from the outstanding feature of the Doc2-Vec model, which can represent a sentence with linguistic, semantic properties and rules that can learn from the context of the sentence.

Moreover, the presentation category achieves the best performance for the RNN (19.50%). On the other hand, the RNN performs poorly in four categories, including peer interactive communication and technical support (5.48%), declaration (8.27%), general acknowledgement (6.07%), and general statement (3.92%). Given that the performance of this classifier is quite different from that of the others, we assume that the RNN is designed to analyze time-series text, which is not in accordance with our training dataset.

Table 8 presents the experimental results on the converted dataset, which follows a pattern similar to that of the original dataset, although the most accurate category is peer interactive

**Table 8. Average of proportional performance results on categories with converted dataset.**

| Algorithms (Accuracy) | Categories | | | | | |
|---|---|---|---|---|---|---|
| | Peer interactive communication and technical support | Declaration | Presentation | Financial | General acknowledgement | General statement |
| CNN (80.00%) | 15.71 | **20.00** | 8.57 | 5.71 | 18.57 | 11.43 |
| RNN (69.03%) | 11.43 | 11.54 | **16.24** | 13.42 | 9.50 | 6.89 |
| CNN+Doc2vec (87.93%) | 20.00 | 14.82 | 11.47 | **20.10** | 10.97 | 10.56 |
| kNN (80.40%) | 19.60 | 12.07 | 11.47 | **20.10** | 7.53 | 9.64 |
| Logistic regression (87.15%) | 20.93 | 14.92 | 11.89 | **21.02** | 8.44 | 9.96 |
| Naïve Bayes (87.20%) | 20.24 | 15.05 | 11.38 | **20.93** | 9.73 | 9.87 |
| N-gram language model (85.87%) | 19.96 | 15.00 | 11.38 | **20.56** | 9.78 | 9.18 |
| *Average of performance per category* | *18.28* | *14.77* | *11.77* | *17.41* | *10.65* | *9.65* |

communication and technical support and the second most accurate one is financial. The peer interactive communication and technical support category achieves 18.28% accuracy on average, followed by the financial category (17.41%). However, the general statement category scores the lowest average (9.65%), which slightly differs from the general acknowledgement (10.65%). Actually, the accuracy of the different classifiers on the two datasets is not significantly different overall. An interesting observation is that the performance of the RNN is improved in almost every category, except for the presentation class, which shows a decrease. In contrast, CNN performance is reduced in many classes, excluding the peer interactive communication and technical support and declaration categories. We assume that the performance is improved because the converted sentences are short and simplified.

Fig 6 represents the performance comparison by the accuracy measures and average of performance per category of each algorithm with the original as well as converted datasets.

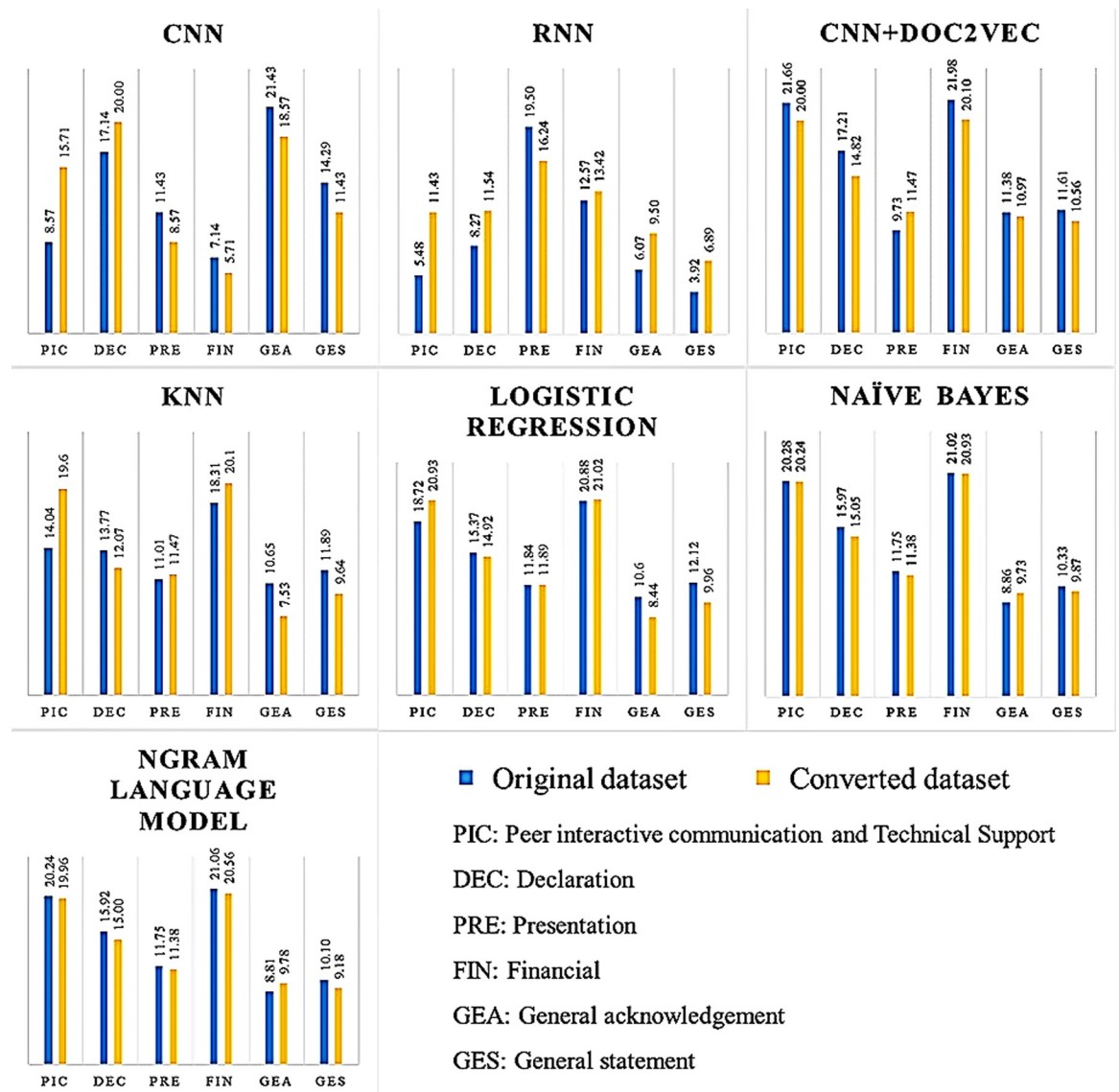

**Fig 6. Performance comparison by accuracy measures on original and converted datasets.**

## Conclusion

In this study, we classified acknowledgements sections automatically with machine learning algorithms including deep learning. We built a sizeable training dataset for automatic classification and investigated the factors affecting the performance of automatic classification.

Throughout the comprehensive examination of factors, we observed that the representation of the training dataset positively influences classification performance, especially of deep learning algorithms. With respect to the effects of acknowledgements sentence and category, we discovered that the length and linguistic and lexical properties of the sentence affected the performance significantly. In particular, converting the training dataset had a big impact on the performance due to its characteristics of being short and lacking contextual terms compared to the original dataset. In addition, in terms of category, some categories such as general acknowledgement and general statement consisted mostly of general terms, which made the performance poor. We also observed that the CNN+Doc2Vec algorithm worked best with the acknowledgements training dataset. It may be attributed to the fact that the acknowledgements section has fixed patterns that are best suited for the CNN with document embedding representation.

This study has the limitation that the acknowledgements were entirely from the biomedical domain. Due to this biomedical domain orientation, the findings reported in this paper may not be applicable to other domains such as the social sciences and humanities. As a follow-up study, we plan to apply our training model to the entire collection downloaded from PMC to understand the characteristics of the acknowledgements section in the biomedical domain. In addition, we plan to develop an online system of acknowledgements classification to enable the wider use of automatic acknowledgements classification.

## Supporting information

**S1 File. The acknowledgement categories.** The categories used in the experiment, also can be found at https://doi.org/10.6084/m9.figshare.11302280.
(PDF)

**S2 File. The characteristics and examples of sentences in each category of acknowledgement.** This information used in the experiment, also can be found at https://doi.org/10.6084/m9.figshare.11302289.
(PDF)

## Acknowledgments

The authors would like to thank you the anonymous reviewers for their insightful comments.

## Author Contributions

**Conceptualization:** Min Song, Keun Young Kang, Tatsawan Timakum.

**Data curation:** Min Song, Keun Young Kang, Tatsawan Timakum.

**Formal analysis:** Min Song, Keun Young Kang, Tatsawan Timakum.

**Funding acquisition:** Min Song.

**Investigation:** Min Song, Keun Young Kang, Tatsawan Timakum, Xinyuan Zhang.

**Methodology:** Min Song, Keun Young Kang.

**Project administration:** Min Song.

**Resources:** Min Song.

**Software:** Min Song.

**Supervision:** Min Song.

**Validation:** Min Song, Keun Young Kang, Xinyuan Zhang.

**Visualization:** Keun Young Kang, Tatsawan Timakum.

**Writing – original draft:** Min Song, Keun Young Kang, Tatsawan Timakum, Xinyuan Zhang.

**Writing – review & editing:** Min Song, Keun Young Kang, Tatsawan Timakum.

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
