## [Decision Letter · Decision Letter 0]

28 Nov 2019

PONE-D-19-22696

Examining Influential Factors for Acknowledgements Classification Using Supervised Learning

PLOS ONE

Dear Author(s),

Thank you for submitting your manuscript to PLOS ONE. After careful consideration, we feel that it has merit but does not fully meet PLOS ONE’s publication criteria as it currently stands. Therefore, we invite you to submit a revised version of the manuscript that addresses the points raised during the review process.

We would appreciate receiving your revised manuscript by Jan 12 2020 11:59PM. To enhance the reproducibility of your results, we recommend that if applicable you deposit your laboratory protocols in protocols.io, where a protocol can be assigned its own identifier (DOI) such that it can be cited independently in the future. For instructions see: http://journals.plos.org/plosone/s/submission-guidelines#loc-laboratory-protocols

We look forward to receiving your revised manuscript.

Kind regards,

Amira M. Idrees, Associate Professor

Academic Editor

PLOS ONE

Journal Requirements:

2. In your data collection section, please include a URL link to the website from which the PMC data was downloaded.

3. We note that your training data was built by human participants.  In the Methods and the ethics statement in the online submission information, please ensure that you have specified (1) whether informed consent was obtained and (2) what type you obtained (for instance, written or verbal, and if verbal, how it was documented and witnessed).

'This work was supported by the National Research Foundation of Korea(NRF) grant funded by the

502 Korea government(MSIT) (No. NRF-2019R1A2C2002577).'

'The funders had no role in study design, data collection and analysis, decision to publish, or preparation of the manuscript.'

Reviewers' comments:

Reviewer's Responses to Questions

**Comments to the Author**

1. Is the manuscript technically sound, and do the data support the conclusions?

Reviewer #1: No

Reviewer #2: Yes

2. Has the statistical analysis been performed appropriately and rigorously? 

Reviewer #1: No

Reviewer #2: Yes

3. Have the authors made all data underlying the findings in their manuscript fully available?

Reviewer #1: Yes

Reviewer #2: Yes

4. Is the manuscript presented in an intelligible fashion and written in standard English?

Reviewer #1: Yes

Reviewer #2: Yes

5. Review Comments to the Author

Reviewer #1: The paper proposed machine learning and deep learning approach for identifying the influential factor of acknowledgement. The research area of this manuscript is interesting for many readers. The presentation of the study is fair and language is understandable.

However, there are major concern about the paper:

The authors required to improve the abstract by adding the aim and method.

What is paper aims?

The paper failed to address the most important requirement of research; which is the current state of the art in the area of bibliometric analysis using machine learning and deep learning approach. Consider identifying the main problem of the proposed studies.

What is paper contribution? Please make clear.

The authors unable to provide clear methodology about how the experiment have been done and analysis the data.

The authors suggested to used third person name than used “we”

The paper organization look more to bibliometric analysis. However, the significant of using machine learning and deep learning on this analysis is not clear.

Most of figure is not clear.

In line 186 section preprocessing, the authors mentioned to separated article to sentences. Why? Please give more explanation about it

In table 3, the authors make a comparison with previous research (Paul-hua), why authors used different name such as funding by Paul to Financial. The authors can follow the previous word. Why you need general statement. Please clarify because it looks more to bibliography.

In step 4, crosschecking with each other. Why authors do not used expert panel to clarify your judgement to the sentences.

How many data training and testing? Not clear

References are not up to date

Reviewer #2: The topic of this paper is interesting and I believe that the authors have made

good and clear contribution.

The contribution was clearly explained .

The presentation of the paper was clearly acceptable and smart.

The English level was quite good.

The results were more than enough since they have provided all the required results for the contribution .

The abstract and conclusion of this paper are concise.

Suggestion:

In the introduction section I suggest to remove the lines 78-82

6. PLOS authors have the option to publish the peer review history of their article (what does this mean?). If published, this will include your full peer review and any attached files.

Reviewer #1: No

Reviewer #2: No

---

## [Author Response · Author response to Decision Letter 0]

9 Dec 2019

Manuscript Journal of PONE-D-19-22696

Examining Influential Factors for Acknowledgements Classification Using Supervised Learning

PLOS ONE

Journal Requirements:

 Thank you for the instruction. We carefully reviewed and revised the manuscript according to your suggestions.

 The necessary modifications have been made to the revised manuscript. The formatting of the revised manuscript is now compliant with the PLOS ONE instructions.

2. In your data collection section, please include a URL link to the website from which the PMC data was downloaded.

 We revised the statement of data collection as follow:

 We downloaded the entire full-text collection from PubMed Central database (PMC) ftp://ftp.ncbi.nlm.nih.gov/pub/pmc.

3. We note that your training data was built by human participants. In the Methods and the ethics statement in the online submission information, please ensure that you have specified (1) whether informed consent was obtained and (2) what type you obtained (for instance, written or verbal, and if verbal, how it was documented and witnessed).

 We added the required information to the Methods section and the ethics statement in the online submission information as follows:

 “Following the principles of informed consent, written informed consent was obtained from each participant.”

 Per your request, we described the change in our cover letter, and we provided two DOIs to access the datasets of the manuscript. We changed our Data Availability statement as follow:

 “The datasets underlying this study are available publicly via two DOIs 1) Acknowledgement training dataset https://doi.org/10.6084/m9.figshare.11321852 and 2) Acknowledgement classification model (Doc2Vec Classifier Model) https://doi.org/10.6084/m9.figshare.11330564”

5. Thank you for stating the following in the Acknowledgments Section of your manuscript: 'This work was supported by the National Research Foundation of Korea(NRF) grant funded by the 502 Korea government(MSIT) (No. NRF-2019R1A2C2002577).'

We note that you have provided funding information that is not currently declared in your Funding Statement. However, funding information should not appear in the Acknowledgments section or other areas of your manuscript. We will only publish funding information present in the Funding Statement section of the online submission form. Please remove any funding-related text from the manuscript and let us know how you would like to update your Funding Statement. Currently, your Funding Statement reads as follows:

'The funders had no role in study design, data collection and analysis, decision to publish, or preparation of the manuscript.'

 We removed the funding-related statement from the Acknowledgements section of the manuscript and rewrote the Acknowledgements section as follows:

 “The authors would like to thank you the anonymous reviewers for their insightful comments.”

 In addition, we updated our Funding Statement of the online submission form as follows:

 'This work was supported by the National Research Foundation of Korea(NRF) grant funded by the 502 Korea government(MSIT) (No. NRF-2019R1A2C2002577). The funders had no role in study design, data collection and analysis, decision to publish, or preparation of the manuscript.'

Review Comments to the Author

Reviewer #1

General comments:

The paper proposed machine learning and deep learning approach for identifying the influential factor of acknowledgement. The research area of this manuscript is interesting for many readers. The presentation of the study is fair and language is understandable. However, there are major concern about the paper:

 Thank you for your review and constructive comments. Taking your comments and suggestions into consideration, we have made requested changes to the manuscript.

Major comments:

1. The authors required to improve the abstract by adding the aim and method.

- What is paper aims?

 Thank you for your comments. We revised the abstract as follows:

 Acknowledegments have been examined as important elements in measuring the contributions to and intellectual debts of a scientific publication. Unlike previous studies that were limited in the scope of analysis and manual examination. The present study aimed to conduct the automatic classification of acknowledgements on a large scale of data. To this end, we first created a training dataset for acknowledgements classification by sampling the acknowledgements sections from the entire PubMed Central database. Second, we adopted various supervised learning algorithms to examine which algorithm performed best in what condition. In addition, we observed the factors affecting classification performance. We investigated the effects of the following three main aspects: classification algorithms, categories, and text representations. The CNN+Doc2Vec algorithm achieved the highest performance of 93.58% accuracy in the original dataset and 87.93% in the converted dataset. The experimental results indicated that the characteristics of categories and sentence patterns influenced the performance of classification. Most of the classifiers performed better on the categories of financial, peer interactive communication, and technical support compared to other classes.

2. The paper failed to address the most important requirement of research; which is the current state of the art in the area of bibliometric analysis using machine learning and deep learning approach. Consider identifying the main problem of the proposed studies.

-What is paper contribution? Please make clear.

 Thank you for your constructive comments. As per your comments, we added the following paragraph to the Introduction section.

 Recently, machine learning as well as deep learning techniques have been widely used for bibliometric research as an effective tool to rank the scientific research [15], predict further citation [16-17], classify citation motivation [18], and author disambiguation [19]. However, these machine learning and deep learning techniques have not been applied to automatic acknowledgements classification at PubMed scale.

 The main goal of the present study was to develop the automatic classification method for the acknowledgements section. In particular, we aimed at two objectives. Firstly, we aimed to create a large-scale training dataset for the automatic classification of acknowledgements. Secondly, we examined which factors affected classification performance. We investigated the effects of the following three aspects: classification algorithms, types of category, and text representations. For classification algorithms, we adopted several supervised learning algorithms, and we compared the performance in terms of accuracy, precision, recall, and F1-score for each classifier. For the effects of categories, we classified our acknowledgements into six categories and examined the characteristics of each category that may affect the performance. For the effects of text representations, we converted a sentence into a simplified format by utilizing several features and applied the Named Entity Recognition (NER) model with Stanford CoreNLP’s seven classes for feature generation to examine the impact of text representations on the performance of classifiers. The present study serves as a core base for analyzing the entire collection of acknowledgements sections from full-text repositories like PMC and understanding the characteristics and patterns of acknowledgements on a large scale. To facilitate research of acknowledgements section, we have made both the training dataset and acknowledgements classification model publicly available at http://informatics.yonsei.ac.kr/acknowledgement_classification/acknowledgement_classification.html

 The main contribution of the present study is to explore the most effective and automatic classification of acknowledgements by selecting salient features for classification and the state-of-the-art algorithms. 

3. The authors unable to provide clear methodology about how the experiment have been done and analysis the data. 

 Thanks for your comments. We revised the manuscript (the Method section) to provide more clear explanations of the methods applied to the experiments as well as analysis of the results as follows: 

 “The methods applied to the experiments consisted of three parts:1) data collection and pre-processing, 2) dataset preparation and transformation, and 3) algorithm implementation and analysis. For data collection and pre-processing, we used the PubMed Central dataset to extract the acknowledgements section. After collecting and pre-processing, we built the training set for classification in two ways. The first training set is comprised of original sentences along with the classification label manually assigned by the experts. The second set consists of the converted sentences by regular expression and Named Entity Recognition (NER) techniques. The reason for providing two different sets of training data is to examine which training dataset works better as features for automatic classification. With these two different training sets, we applied various deep learning and machine learning algorithms to measure the performance of classification of acknowledgement sentences. In the analysis phase, we examine the impact of classification algorithms as well as the types of text representation on the performance of classification.”

4. The authors suggested to used third person name than used “we”

 Thank you for your comments. We consulted with the English professional editor about this. The editor referred to the following rules:

 APA (The American Psychology Association) has the following to say about the use of "we" (p. 69-70).

 To avoid ambiguity, use a personal pronoun rather than the third person when describing steps taken in your experiment. Therefore, in our case, using “we” is appropriate.

5. The paper organization look more to bibliometric analysis. However, the significant of using machine learning and deep learning on this analysis is not clear.

 Thank you for your comments. 

 The reason that the manuscript looks more like bibliometric analysis is because we started with the problems of bibliometric analysis and in fact, the acknowledgements section is one of the promising but relatively unexplored research area in bibliometrics. However, after modifying the Introduction section to include machine learning and deep learning techniques in bibliometric analysis, we believe that the significance of machine learning and deep learning in analysis of acknowledgements became more highlighted. 

6. Most of figure is not clear.

 Thank you for your comments. 

 We adjusted the resolution of the figures to be 300 dpi and they are more clear now.

7. In line 186 section preprocessing, the authors mentioned to separated article to sentences. Why? Please give more explanation about it.

 Thank you for your comments. 

 We gave more explanations about splitting the sentence in the data processing section as follow:

 “The reason for separating an acknowledgements section into sentences is because there could be multiple reasons for acknowledging someone or some entities. Thus, to analyze types of acknowledgements at a fine-grained level, we decided to split the acknowledgements section into sentences.” 

8. In table 3, the authors make a comparison with previous research (Paul-hua), why authors used different name such as funding by Paul to Financial. The authors can follow the previous word. Why you need general statement. Please clarify because it looks more to bibliography.

 Thank you for your comments.

 To name the categories of acknowledgements, we considered the content and schemes based on previous studies (Table 1) to categorize acknowledgements into six categories (Table 2). We marked some categories differently from the prior studies to cover the content of the data that we classified, such as a category of “Presentation.” We described the content of each category in Table 2. For some categories, we borrowed the class labels from previous studies such as “Financial.”

 With regard to the study of Paul-Hua et al. (Table 3), we used the keywords and the usage patterns provided in their paper as a guideline to determine the categories of acknowledgements. In addition, we expanded the keywords and usage patterns from our datasets to provide more accurate training dataset.

 In the account of your useful comments, we have made the changes to the description of Table 2 and Table 3 as follows:

 To classify the sentences of acknowledgement, we considered the content, language patterns, and schemes based on previous studies and the characteristics of our raw data of acknowledgements. In this study, we divided the classes of acknowledgements into six categories, which are shown in Table 2.

Table 2. Categories of acknowledgements in this study.

 We devised new class labels that were not introduced in the prior studies to cover the content of the data that we classified, such as a category of “Presentation” and “Peer interactive communication and technical support”. For some categories, we still borrowed the class labels from the previous studies such as “Financial” [20, 22, 25-26].

 To determine the categories of acknowledgements, we used the keywords and their usage patterns in the acknowledgements section. A similar approach was adopted in the study of Paul-Hua et al. [41] that we used as a guideline to categorize the acknowledgements sentences. They extracted acknowledgements indexed in the WoS, SCI-E, and SSCI to identify the types of acknowledgements. They applied linguistic pre-processing tools, such as a tokenizer and POS tagger, and modified the grammatical rule set for noun phrase chunking to obtain a result. We took this result to consider our classification as well. For example, to consider our class of Peer interactive communication and technical support, related terms such as discussion, advice, guidance, image analysis, and laboratory tools were observed for classification. The summary of noun phrases is shown in Table 3.

Table 3. Examples of noun phrase patterns in each type of acknowledgements category.

9. In step 4, crosschecking with each other. Why authors do not used expert panel to clarify your to the sentences.

 Thank you for your comments.

 In step 4, “crosschecking with each other,” the researchers who are Ph.D. candidates, majored in the library and information science, curated the training dataset. These three researchers claimed to be experienced and skillful researchers. Therefore, with the thorough cross-check process, we feel confident that our judgment for classifying the acknowledgements sentences is accurate. 

 To make this step to more clearly understood, we added more explanations about the crosscheck process as follows:

 “Before making the final version of the training set, we performed a secondary step for checking the data. Three researchers who are Ph.D. students that majored in the library and information science crosschecked all of the data with the guidelines. For this step, we made agreements and calculated the agreement rate as follows:

(1) If all three researchers classified the sentence into the same category, the agreement rate was 100%. 

(2) If two researchers classified it into the same category and one was different, the rate was ~ 66.67%. 

(3) If all researchers classified it differently but discussed and decided to focus on one category, the rate was 0%.

According to these rules, we made the final version of the training dataset with an average agreement rate of 97.27%, and the final number of sentences for each category, shown in Table 4, was 1,815 on average.”

10. How many data training and testing? Not clear.

 Thank you for your comments. 

 The training dataset consists of 10,893 sentences. Like in the evaluation of any machine learning or deep learning algorithms, we split the training dataset into two sets at 80:20 rate for the training and the validation set, respectively. 

 We also added this information in the Building training data section.

11. References are not up to date

 Thank you for your comments.

 This study is related to acknowledgements analysis, which has a long history of research in bibliometrics. Most of the previous studies in this line of research were manual classification with a handful size of data and were conducted several decades ago. To justify our decision of the categories of aknowldgements, we cited those previous studies. In addition, we added more recent references to the Introduction section for applications of machine learning and deep learning to bibliometric analysis. 

Reviewer #2

General comments:

The topic of this paper is interesting and I believe that the authors have made good and clear contribution.

- The contribution was clearly explained.

- The presentation of the paper was clearly 

 acceptable and smart.

- The English level was quite good.

- The results were more than enough since they 

 have provided all the required results for the 

 contribution.

- The abstract and conclusion of this paper are 

 concise.

 Thank you for your encouragement and positive comments. 

Major comments:

In the introduction section I suggest to remove the lines 78-82

 Thank you for the useful comment. 

 We revised the manuscript according to your suggestions. In the Introduction section, the line 78-82 were deleted.

---

## [Decision Letter · Decision Letter 1]

28 Jan 2020

Examining Influential Factors for Acknowledgements Classification Using Supervised Learning

PONE-D-19-22696R1

Dear Author(s),

We are pleased to inform you that your manuscript has been judged scientifically suitable for publication and will be formally accepted for publication once it complies with all outstanding technical requirements.

With kind regards,

Amira M. Idrees, Associate Professor

Academic Editor

PLOS ONE

Additional Editor Comments (optional):

Reviewers' comments:

Reviewer's Responses to Questions

**Comments to the Author**

1. If the authors have adequately addressed your comments raised in a previous round of review and you feel that this manuscript is now acceptable for publication, you may indicate that here to bypass the “Comments to the Author” section, enter your conflict of interest statement in the “Confidential to Editor” section, and submit your "Accept" recommendation.

Reviewer #1: All comments have been addressed

Reviewer #2: All comments have been addressed

2. Is the manuscript technically sound, and do the data support the conclusions?

Reviewer #1: Yes

Reviewer #2: Yes

3. Has the statistical analysis been performed appropriately and rigorously? 

Reviewer #1: Yes

Reviewer #2: Yes

4. Have the authors made all data underlying the findings in their manuscript fully available?

Reviewer #1: Yes

Reviewer #2: Yes

5. Is the manuscript presented in an intelligible fashion and written in standard English?

Reviewer #1: Yes

Reviewer #2: Yes

6. Review Comments to the Author

Reviewer #1: All comments have been addressed well. The authors answers all the comments and make amendment in their paper.

Reviewer #2: (No Response)

7. PLOS authors have the option to publish the peer review history of their article (what does this mean?). If published, this will include your full peer review and any attached files.

Reviewer #1: No

Reviewer #2: No

---

## [Editor Report · Acceptance letter]

31 Jan 2020

PONE-D-19-22696R1 

Examining Influential Factors for Acknowledgements Classification Using Supervised Learning 

Dear Dr. Song:

I am pleased to inform you that your manuscript has been deemed suitable for publication in PLOS ONE. Congratulations! Your manuscript is now with our production department. 

With kind regards,

on behalf of

Prof. Amira M. Idrees 

Academic Editor

PLOS ONE